# Why Do People with Severe Mental Illness Have Poor Cardiovascular Health?—The Need for Implementing a Recovery-Based Self-Management Approach

**DOI:** 10.3390/ijerph182312556

**Published:** 2021-11-29

**Authors:** Sara Zabeen, Sharon Lawn, Anthony Venning, Kate Fairweather

**Affiliations:** 1College of Medicine and Public Health, Flinders University, Adelaide 5042, Australia; sharon.lawn@flinders.edu.au (S.L.); anthony.venning@flinders.edu.au (A.V.); kate.fairweather@flinders.edu.au (K.F.); 2Menzies School of Health Research, Charles Darwin University, Darwin 0811, Australia

**Keywords:** severe mental illness, cardiovascular disease, comorbidity, chronic condition self-management, recovery, caregiver, self-care, integrated care

## Abstract

People with severe mental illness (SMI) die significantly earlier than their well counterparts, mainly due to preventable chronic conditions such as cardiovascular disease (CVD). Based on the existing research, this perspective paper summarises the key contributors to CVD in people with SMI to better target the areas that require more attention to reduce, and ultimately resolve this health inequity. We discuss five broad factors that, according to current international evidence, are believed to be implicated in the development and maintenance of CVD in people with SMI: (1) bio-psychological and lifestyle-related factors; (2) socio-environmental factors; (3) health system-related factors; (4) service culture and practice-related factors; and (5) research-related gaps on how to improve the cardiovascular health of those with SMI. This perspective paper identifies that CVD in people with SMI is a multi-faceted problem involving a range of risk factors. Furthermore, existing chronic care or clinical recovery models alone are insufficient to address this complex problem, and none of these models have identified the significant roles that family caregivers play in improving a person’s self-management behaviours. A new framework is proposed to resolve this complex health issue that warrants a collaborative approach within and between different health and social care sectors.

## 1. Introduction

In Australia and internationally, adults with SMI such as schizophrenia, schizoaffective disorder, and bipolar disorder are at increased risk of early death due to physical health problems. Cardiovascular diseases (CVD), type 2 diabetes, nutritional and metabolic diseases, and obesity and smoking-related cancers are some of these potentially preventable chronic conditions that are frequently observed and disproportionately found among people with SMI compared with non-SMI populations [1]. This mental–physical health interface has multiple adverse impacts for the person in their daily life, and also appears to generate greater difficulty in the person’s healthcare management. It is particularly challenging for community-living adults with SMI who have to rely on the overburdened and often fragmented public health and support service systems, resulting in poorer health outcomes [2]. Overall, an estimated AUD 15 billion is spent across Australia and New Zealand each year on healthcare costs related to people with SMI experiencing comorbidities and resultant mortality [3], where a significant proportion of deaths involve CVD [1]. Accordingly, the economic and social burden of SMI-related CVD on the healthcare system and the community is a major cost to all key stakeholders (e.g., persons with SMI, their families, health practitioners, policy makers) [3].

Despite a large body of research documenting the SMI and physical comorbidities relationship and efforts to meet the complex medical needs of the population with SMI, addressing this health inequity remains at the periphery of research and healthcare practice [4]. These physical health disparities in people with SMI are observed across low-income, middle-income and high-income countries, making it a multifaceted global health problem [5,6]. Through a meta-analysis of 100 systematic review papers, Firth et al. (2019) showed that, globally, people with SMI are 1.4–2.0 times more likely to have CVD [5].

The following section aims to discuss key drivers contributing and perpetuating this status quo over decades. Synthesising the findings involving these key drivers will help to fulfil the objective of this study, which is to propose a framework suitable to better understand and tackle this important health problem. It is anticipated that by identifying these critical issues, readers may better understand the focal points at the individual, health system and socio-environmental levels, with potential to catalyse improvements in the cardiovascular health of those with SMI.

## 2. Methods

The perspectives on how to improve the cardiovascular health of those with SMI discussed in this paper are drawn from a range of sources. Given that the literature on physical health and mental health is vast, we undertook a more targeted approach in order to identify papers based on their perceived and reported reach, influence and importance in this area of research. We used the Google Scholar search engine, using the phrase ‘cardiovascular disease in people with severe mental illness’ to identify systematic reviews and peer-reviewed articles that discussed the broader socio-environmental, cultural and political factors impeding the cardiovascular health of those with SMI. We limited our search to papers published since 2000 because there was evidence of a clear shift from that time, with increasing calls to address this topic. In Australia, for example, the first national report on physical health and mental health (The Duty of Care Report), linking several national datasets for the first time, was released two decades ago, elevating the significance of this topic nationally [7]. We also identified reports from government bodies and nongovernment organisations (e.g., The World Health Organization), professional bodies (e.g., The Royal College of Psychiatrists) and advocacy organisations (e.g., The King’s Fund) on this topic from Australia, the United Kingdom, the United States and other jurisdictions that have been developed since 2000. These were identified similarly through internet searching, and also drawing on the knowledge and expertise of the first author’s supervisory team. Further papers were identified by checking cross-references of the systematic review articles. Following a series of discussions between the first and second authors, 48 papers were included and synthesised in the following section to better understand why people with SMI experience a greater prevalence of CVD and related risk factors.

## 3. Chief Contributors to CVD among People with SMI

Health research investigating the excess of CVD-related mortality associated with SMI reveals multi-level factors that exacerbate poor heart health. Existing evidence can be divided into five key areas, linked to the marginalisation of people with SMI: (1) bio-psychological and lifestyle-related risk factors; (2) socio-environmental factors; (3) health system-related factors; (4) service culture and practice-related factors; and (5) research-related gaps on how to improve the cardiovascular health for people with SMI.

### 3.1. Bio-Psychological and Lifestyle-Related Factors Impacting Cardiovascular Health of People with SMI

Coping mechanisms such as tobacco smoking, excessive alcohol consumption, poor diet, and physical inactivity are common behaviours among people with SMI [5], many who consider these a consequence of trying to manage their mental illness. These risky behaviours contribute to the high rates of chronic physical illnesses, the most common being CVD [1]. A clear association between SMI and CVD identifies that the comorbidity of these chronic conditions has a shared biological (older age, male gender, family history), psychological (stress, major depression) and lifestyle (smoking, alcoholism, poor diet, lack of exercise) aetiology [8]. Further, a person with SMI typically experiences a combination of three main symptom types across the course of their illness: positive (delusions and hallucinations), negative (amotivation and anhedonia) and cognitive symptoms (poor concentration, reasoning and memory) [9]. The influence of these symptoms and their interaction severely impact the day-to-day activities of people with SMI, making it difficult for them to manage their mental and physical health effectively.

### 3.2. Socio-Environmental Factors Impacting Cardiovascular Health of People with SMI

Besides the bio-psychological and lifestyle-related risk factors linking SMI and CVD, common socio-environmental factors such as financial hardship, lack of access to community resources, and social isolation appear to perpetuate the comorbidity of these conditions [10]. Researchers have observed that the nature and complexity of the SMI-related symptoms (disorganised behaviours, major depression and anxiety, cognitive impairment, problems with social skills) often lead to high rates of unemployment, underpinning the financial hardship among people with SMI [11]. Financial hardship frequently impacts this group’s capacity to access transportation and maintain consistent housing or accommodation [12]. The unpredictability or absence of housing and transport can disengage individuals with SMI from their community, diminishing their overall quality of life.

Social support is another crucial protective factor known to improve the overall health of people, especially those living with mental–physical health comorbidity [4]. Due to the complexity of SMI and related positive symptoms (delusions, hallucinations), it is not unusual for this population to experience significant social stigma, self-stigma, social exclusion and isolation [13]. As a consequence, people with SMI may be more prone to live in unsafe environments that expose them to greater risks of substance abuse such as heavy smoking, alcoholism and drug misuse [11,12], contributing to the likelihood of CVD. In addition, due to the ongoing social stigma towards mental illness, service users may struggle to fully engage and participate in, or gain equitable access to, meaningful community activities as citizens. The combination of the aforementioned factors, the debilitating nature of SMI and the potential low physical health literacy, has created further challenges for this population in managing their health conditions [14]. Ultimately, these circumstances are implicated in reducing self-esteem, negatively impacting their treatment-seeking, and delaying the delivery of effective prevention, early intervention and treatment for their health conditions [15].

### 3.3. Health System and Practice-Related Factors Impacting Cardiovascular Health of People with SMI

Health system-centric factors increasing the risk of CVD among those with SMI include inconsistency in the provision of quality medical care tailored for this population group. For instance, General Practitioners (GPs) are well-positioned to promote healthy lifestyles in people with SMI [11]. However, evidence suggests that feeling stigmatised in a primary health care setting, coupled with GPs’ lack of experiences in managing people with SMI, is likely to undermine help-seeking behaviours and adherence to treatment plans recommended by GPs [13]. Consequently, some researchers have argued that physical healthcare should be the responsibility of mental health professionals (particularly psychiatrists and mental health nurses), owing to their existing expertise with this group [16]. However, both internationally and in Australia, the major shift of psychiatry towards ‘deinstitutionalisation’ and ongoing reforms have overburdened the mental health system [2]. Similarly, structural changes in primary and secondary care have led to increased levels of bureaucracy and increased pressure on already scarce health resources (insufficient skilled staff and time), creating further challenges for SMI populations to access GPs’ services [10,17]. There are ongoing dilemmas regarding *who* is well-suited and responsible for managing the physical health needs of people with SMI [18]. Furthermore, where there are clear guidelines on how to identify and manage physical health risk factors among people with SMI, health policies and procedures are poorly implemented; here, Australia is no exception [18]. As a consequence, people with SMI have limited access to preventative care, motivational interventions and evidence-based guideline-concordant treatments that address both lifestyle risk-factors and physical illnesses such as CVD [19].

### 3.4. Service Culture and Policy-Related Gaps on How to Improve the Cardiovascular Health of People with SMI

Although there are strong signals identifying an association between CVD and SMI risk factors, person-centred evidence-based healthcare (EBHC) interventions relating to CVD-risk reduction strategies are rare [20]. Recovery-oriented and self-management-based interventions, tailored information, education and training incorporating community support are considered crucial for potentiating physical health for this population [21]. Research has indicated that, when these interventions are implemented, better health outcomes are observed owing to the approach enhancing individuals’ abilities, knowledge and confidence to manage their conditions [22]. Furthermore, the provision of a person-held medical record has been proven to improve the knowledge of physical health risks among those with SMI [23]. A person-centred approach to healthcare acknowledges the important contributions of the families of people with SMI to care decisions and support, and emphasises partnership and collaborative interventions. Peer support, and clear and brief health promotion messages are well suited to this population [24]. In practice, however, most of these elements are overshadowed within predominantly hierarchical healthcare systems primarily interested in maintaining acute care, risk and medication compliance, with least focus on the long-term holistic psychosocial wellbeing and needs of this population [25,26]. This ‘illness’ focus is notably at odds with a recovery-orientation [27,28].

Health service culture can also negatively impact the uptake of recovery-oriented and self-management-based physical health interventions specific to the SMI population [27]. Broadly, among mental health organisations, there is a service culture perception that improving physical health behaviours among SMI persons is highly challenging [23]. This cultural inertia includes the acceptance of obesity as an *inevitable* side-effect of antipsychotic medications [29]. Similarly, smoking by people with SMI continues to be considered as an embedded culture or group norm within mental health services [30]. Thus, pervasive pessimism regarding the capacity of people with SMI to embrace health behaviour change contributes to the failure to treat these common CVD risk factors among people with SMI [31]. Nevertheless, evidence shows that changes in mental health organisational culture can be accomplished with targeted improvement strategies [32]. Such improvement strategies targeting culture can concentrate on the major impeding factors, such as an absence of strong clinical leadership and engagement, and a lack of coordination in inter-organisational partnerships and teamwork [33]. However, the implementation of the aforementioned approaches meets consistent resistance from the mental health system, as novel strategies are often considered a threat to extant rules and regulations; thus, both fail to be initiated and subsequently sustained [33].

Beyond health system-level cultural issues, models of practices (e.g., individually based illness model vs. recovery approach) also influence treatment methodologies, and are seen as problematic among mental health researchers [27,34]. For instance, to date, psychiatry heavily relies on a medication-centric model, although research has repeatedly showed the adverse impacts of a variety of these drugs on people’s overall health and wellbeing [35]. It is also important to note that some of the CVD-related risk factors themselves may be the ‘iatrogenic (inadvertently induced and often harmful) consequences of antipsychotic medications’ [36]. These include, for instance, amplified appetite that is driven by some of the antipsychotic medications, resulting in negative impacts on metabolism and rapid weight gain [35]. Sedation and other common side-effects of antipsychotic medications can also hinder the person’s motivation and energy to make healthy lifestyle choices [26]. Adding further complexity, for some people with SMI, these antipsychotic medications are ‘imposed coercively’, with limited collaborative decision-making with the person about medication choice and dose, as part of them being detained for in-person treatment, or as part of compulsory Community Treatment Orders (CTOs) [34]. As Laugharne and Priebe [37] found, this issue of power imbalance between persons and clinicians was identified as a contributor to people with SMI disengaging from mental health services. A recent Cochrane review found that CTOs did not result in improved service use, social functioning or quality of life compared to voluntary care [38]. However, the authors acknowledged that the findings were inconclusive due to the small number of articles. Despite the findings, it is important to note that coerced treatment inhibits active engagement by the person in their care [39], and damages their trust in healthcare providers [40], known to be crucial in improving overall health outcomes [41]. The provision of correct and sufficiently communicated information, treatment choice, continuity of care and a consistent attending clinician are deemed critical components to enable people to retain trust in mental health service systems [37,42].

### 3.5. Research-Related Gaps on How to Improve Cardiovascular Health of People with SMI

The acknowledgement of the acceptability and potential clinical effectiveness of integrated general medical and psychiatric care interventions has led to policymakers recently reorienting towards the adoption of recovery-oriented EBHC programmes within the mental health system [13]. However, Hannigan and Coffey [43] noted a paradox: the act of such an implementation has created further challenges by initiating a variety of top-down interventions without considering how delivery will be achieved in different contexts. Similarly, another major disconnection is between the research sector and community health practices, resulting in the poor translation of EBHC within real-world settings [44]. To ensure effective physical health outcomes in people with SMI, research will need to align with policy priorities that are fully informed by the practice field. To achieve this alignment, research and policy makers will need to understand the practical challenges faced by frontline clinicians such as psychiatrists, mental health nurses, and GPs [13]. Of further concern is that current EBHC research is predominantly focused on ‘biomedical’ self-management and related individualistic factors. Biomedical self-management involves *both* ‘work’ and ‘cost’ for persons, where the management burden is often shifted from the health care system to the person [22,45]. This contrasts with ‘lifeworld’ self-management and related research, which additionally considers socio-environmental factors, where the problems originate and are sustained [45].

Existing research also suggests that outcome evaluation alone is generally inadequate to address *wicked problems* (problems that are often resistant to change or improvement initiatives) [46], as with those highlighted in this review. A naturalistic study investigating health settings from both ‘inside’ and ‘outside’ is needed to examine the lifeworld implementation of EBHC and self-management-based interventions in community mental health settings [45]. Few studies have represented a holistic picture of the factors that impact and shape the uptake of chronic condition self-management (CCSM) behaviours and related health practices [45,47]. While progress may be made by employing person-centred physical health interventions [44], it often fails to achieve replicability and sustainability due to the poor understanding of underlying processes facilitating or impeding the delivery of these interventions in the first place [48]. The status quo reflects the clear paucity of research into the successful implementation of person-centred physical health interventions among people with SMI.

## 4. Discussion

From the above articulated concerns, it is apparent that there remain identifiable CVD risk factors for people with SMI at the individual, community, health system and culture, and research level [49]. There is no doubt that this is a ‘complex health problem’ and, therefore, needs a ‘complex solution’ [50]. However, there is ‘no quick fix’, and systematic changes with long-term patient-centred interventions will be required for sustainable outcomes [25,48]. Globally, CCSM-based physical health interventions among people with SMI have shown promise in supporting a person’s overall health and wellbeing [44,48,51,52]. The underlying mechanisms of CCSM aim to enhance the person’s self-efficacy so that, with the help of their support network, they can better manage their own health conditions [47]. Researchers have identified good evidence for CCSM-based tailored weight-loss programmes and mixed-evidence for smoking cessation, substance abuse and risky sexual behaviour in this population [49].

There is also evidence for recovery-oriented models in improving the wellbeing of people with SMI [53,54]. Leamy’s personal recovery framework CHIME (Connectedness, Hope and optimism, Identity, Meaning, Empowerment) is one such model [55]. However, recent research has argued that such models cannot address structural disadvantages or social marginalisation that a person with SMI might face [56]. Current models also fail to capture the organisational barriers that need addressing in order to promote recovery-oriented care. Furthermore, none of the existing models identify the caregiver roles in promoting recovery and improved self-management behaviours in service users [57]. Therefore, the following section proposes a new framework that has elements of different chronic disease models [58,59], recovery [55] and social justice [60] models (Figure 1: A recovery-based self-management framework). This framework has the scope to promote recovery-oriented care in people with SMI and CVD (plus other similar chronic diseases such as type 2 diabetes) risk factors.

Based on the areas identified and discussed in this perspective paper, this framework identifies the key areas that require attention when implying CCSM and recovery-based care in a health setting.

### 4.1. Self-Management

When the word self-management is used in this framework, it is referring to the ‘supported self-management’ (SSM) care where the person is actively supported by their care coordinator [61]. In SSM care, the service user is not only given information, but their care coordinator also helps them achieve their personalised goals by active follow-ups and linking service users with community resources. It is also important that the service user’s cultural and religious beliefs are respected and considered when providing such care. This practice is particularly important for a multicultural country such as Australia. Besides, people with SMI are already a vulnerable cohort who often do not seek active care from the health system [37,42]. Therefore, it is important to create a safe environment for this patient cohort where they are comfortable to seek care and empower themselves. Above all, it is important that the person with SMI is encouraged to be more proactive in making decisions regarding their own treatment—current models seem to overlook this aspect of the SSM care and, hence, do not assert clear roles service users should be playing in a patient-centred care. Addressing these issues will improve the practice-related gaps in the health system by defining both health professional and service user roles. Connecting service users with appropriate community resources will also improve their psychosocial support that can ensure sustainability of healthy behaviours.

### 4.2. Recovery

Personal recovery is another crucial arm of this proposed new framework. There is no doubt that the elements of CHIME [55] and other recovery models [54] play significant roles in motivating a person with SMI to initiate self-management and self-care. The same reasons also help sustain improved health behaviours in people with SMI. However, the existing literature fails to justify the importance of families’ roles in a person’s recovery and self-management journey in achieving improved heart health [57]. Hence, we propose to include this dimension of care into the recovery-based self-management framework. Caregivers’ inside knowledge about service users is useful for the health system. Giving hope to service users, helping them build positive self-identities and promote self-esteem are some of the common outcomes of caregiver-supported recovery. However, it is also reported that caregivers might often need to accompany the person with SMI in the adoption of a new healthy behaviour such as exercise, giving up smoking or alcohol, and eating healthy [57]. By acknowledging and optimising this crucial support that caregivers provide in service users’ lives, care coordinators can hope to achieve more meaningful physical health outcomes in people with SMI and CVD risk factors. The active involvement of caregivers in people’s treatment plan can equip them with the necessary skills to support service users to address deteriorated mental and physical health symptoms, which are important steps that should be incorporated within the current health practices. Efficient use of caregivers’ experience, knowledge and support has the potential to reduce the burden on the health system as well. In addition, the changed health behaviours in service users can bring positivity into both parties’ lives and can make their relationship stronger [62].

Another important part of recovery is connectedness—at individual, social and spiritual levels. Iwasaki et al. [63] explained how therapeutic and general leisure activities can help people with SMI to engage in meaningful community activities. In their study, Iwasaki et al. [63] showed the significance of people with SMI engaging in creative activities such as art, crafts, music, and reading, because engagement with these things, in turn, activates positive emotions. Moreover, the authors also emphasised the importance of activities such as bush-walking, camping, hiking, and individual and team-based sports that were identified as ‘meaning-making’ and activities that ‘heal connections’. This study also highlighted the importance of religious institutions for fostering spiritual connection and a sense of community for some individuals that improved motivation towards recovery and self-care [63]. Hence, researchers, health professionals and caregivers should further explore how these activities can create the conditions for supporting people with SMI and CVD risk factors.

### 4.3. Positive Policy Environment

To date, psychiatry has heavily relied on an illness model where the service user is treated for their illness with medications mostly, and this has been argued to generate iatrogenic trauma for many patients [34]. According to health professionals, high workload, frequent staff turnover and resultant discontinued care, role ambiguities between mental and physical health staff and a lack of resources are some of the other commonly identified barriers that hinder the providing of quality service and recovery-oriented care to service users [18,19]. Recovery-oriented care demands time and is a risk-preferred model, which often becomes difficult to adopt within a community mental health system that is resource-scarce [34]. From a service culture viewpoint, coercive and controlling roles arising from CTOs might also need further consideration regarding how to make these practices more humane, least restrictive, and acceptable to service users. Hence, this proposed framework identified some key areas that should be in place before trying to promote a self-management intervention to improve the cardiovascular health of those with SMI. These areas are: (a) support staff with training, job security and emotional assistance (the emotional support is required to deal with vicarious trauma caused due to service users’ illness), (b) changed paradigm (move towards a recovery-oriented model and away from an illness model where health professionals may hold negative preconceptions regarding the service users’ scope of and capacity for recovery), (c) align sectoral policies (such as better coordination between mental health nurses, psychiatrists and rehabilitation staff), (d) managing political environment (to prevent top-down, unplanned interventions without staff consultation), and (e) better liaising with the physical health staff by sharing real-time medical records through a shared database, where these areas are important to promote effective self-management care to service users [5,31,32,58]. Effective leadership is also key here to promote and sustain the cultural shift needed within psychiatry to ensure the service users’ perspectives are maximised in decisions about treatment and care, and recovery orientation [45]. Most recently in Australia, a large collaborative network of stakeholders, known as Equally Well, has formulated a detailed set of recommendations for policy to address the physical health of people with mental health issues [64]. Paying attention to these issues can improve the service culture and health practices of psychiatry and address the broader structures of healthcare delivery and shared care, to create a positive health model that promotes wellbeing.

### 4.4. Social Justice

Aside the above three elements of the framework, it is important to identify and tackle the root causes such as discrimination, social stigma and resultant isolation, unemployment-generated poverty, and social inequality that perpetuate poor health conditions in people with SMI. Recent research has demonstrated that personal recovery models do not reflect these broader social structural problems [53,56,65], and as a result, fail to achieve sustainable outcomes. Trauma and iatrogenic trauma are other dismaying causes that require close attention if we want genuine changes in service users’ health behaviours [66]. Ensuring these basic rights can help service users be viewed as citizens who can actively contribute to the society, with a broader focus on the social determinants of recovery.

The overall findings suggested that addressing the policy and health-system level factors such as embracing a social recovery-oriented approach (as opposed to clinical recovery) is the key in addressing the long-lasting health problem of CVD in people with SMI. Caregivers should also be engaged more actively and systematically in a person’s treatment plan (where applicable and with the person’s consent)—this will be beneficial for all stakeholders and, as such, can reduce the burden on the resource-limited health system.

## 5. Study Strengths and Limitations

This paper has highlighted the factors that are common and impact people with SMI across the full spectrum of social classes—that is, the study findings and generated framework are applicable to all countries regardless of whether they are low-income, middle-income or high-income countries. The framework is also suitable to other chronic diseases such as type 2 diabetes and obesity that are commonly observed in people with SMI. A further strength of the proposed model is that it highlights caregivers’ roles towards service users’ recovery and improved self-management behaviours, meeting the need of a health system that often fails to duly appreciate and utilise caregivers’ knowledge and expertise in supporting self-management care. To the authors’ best knowledge, no other chronic disease model explicitly identifies and portrays caregivers’ roles in service users’ overall health and wellbeing [57,59].

This study has a number of limitations that need to be acknowledged. The study is not a systematic review and, therefore, some important studies in the literature might have been excluded. A perspective approach to our review was chosen because it met the needs and aims of the study to present a broad discussion to key areas from the peer and grey literature. However, a more systematic approach to searching and synthesis, including appraisal of selected paper quality, is worthwhile.

There are likely areas that require further exploration to better understand the discussed study topic. The authors intend to further refine the framework with their ongoing research on the topic. In the near future, primary qualitative data obtained from all key stakeholders and a realistic literature review will help fulfil this goal.

## 6. Conclusions

From this brief discussion, it is irrefutable that there remain identifiable CVD risk factors for people with SMI spanning individual, community, health system and culture, and research levels [49]. It is argued that mental health issues are a major risk factor for CVD, both directly (i.e., genetic vulnerabilities) and indirectly (e.g., from the lack of access to proper physical healthcare, side-effects of psychotropic medications), impacting the recovery capacity for people with SMI [25,26,65,66]. The adoption of patient-centred EBHC by mental health services would greatly improve the quality of primary care services and consumer outcomes [67]. A wide variety of interventions have been used to improve preventive health management services through mental and physical health service integration and by supporting persons with SMI to change their health behaviours [44,68]. However, few studies have measured outcomes directly relating to CVD risk using recovery and self-management-based approaches [44,49]. To address this gap in the current integrated care and implementation research, the proposed framework has employed key elements of different chronic disease, recovery, and social justice models. This process has helped fulfil the study objective by proposing a comprehensive health model that has the potential to tackle CVD risk factors in people with SMI. According to this framework, when all key stakeholders acknowledge their roles and are actively engaged in the health system and in the community, there is a greater chance of achieving improved cardiovascular health among people with SMI. This paper asserts that CCSM is not a solo pursuit but shared by a distributed network of formal and informal community support providers who require the information and skills needed to effectively support people with SMI. This support network provides the foundation for people with SMI to improve self-care and the capacity to live a meaningful life [65,66]. Similarly at the health system level, more skilled staff and resources, collaborative and integrated care, addressing adversity across the life course as an environmental risk factor, and the provision of more tailored psychosocial supports are required to improve the cardiovascular health of people with SMI [69,70]. There is ‘no quick fix’ available to resolve this complex and *wicked* health problem; systematic changes with long-term person-centred interventions will be required for sustainable outcomes [70,71].

## Figures and Tables

**Figure 1 ijerph-18-12556-f001:**
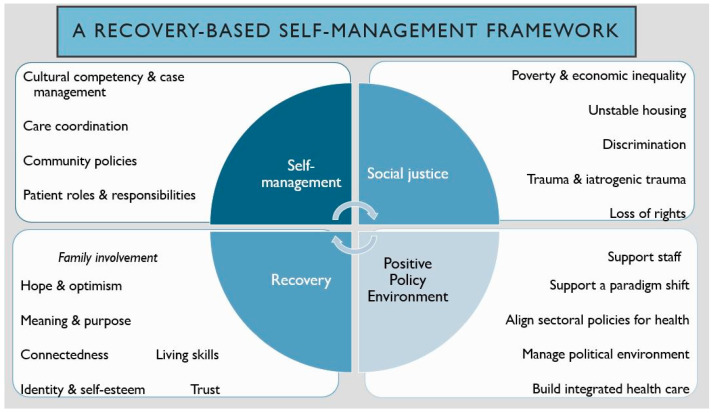
A recovery-based self-management framework.

## Data Availability

Not applicable (a perspective paper).

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
