# Peer review of "Why Do People with Severe Mental Illness Have Poor Cardiovascular Health?—The Need for Implementing a Recovery-Based Self-Management Approach"

_ijerph, 2021, doi:10.3390/ijerph182312556_

Round 1

Reviewer 1 Report

Dear authors, in the attached document you can find the considerations and suggestions for improving your work.
Best regards.

Reviewer 2 Report

Introduction
  Present data in developing or underdeveloped countries so that the application of approaches are global
How do you practice leisure activities? Culture? Sports in this audience? I think that addressing these aspects will also elucidate the issue
no objective was set in the introduction, do it

Methods
Include an item with the description and type of study, the review method, the databases studied, the study selection criteria

Discussion:
Consider the same points from the introduction to enrich the discussion

Conclusion:
As no objective was established in the introduction, it is difficult to evaluate this item. After including the study objective, answer the objective in the conclusion.

Reviewer 3 Report

The paper presents as a synthesis of previous models an integrative approach in the context of mental health, specifically in relation to people suffering from SMI. As a theoretical approach it is an interesting starting point that would require a more empirical next step aimed at validating the claims that are made in this work. I would therefore recommend that the authors indicate this aspect in some way at the end of the document, perhaps as a future line of research.

Round 2

Reviewer 1 Report

Dear authors, I thank you for taking my suggestions into account and I congratulate you because with the changes applied, I believe that your work has improved remarkably. Best regards.